# Non-Uniform Synthetic Aperture Radiometer Image Reconstruction Based on Deep Convolutional Neural Network

Chengwang Xiao [1], Xi Wang [2], Haofeng Dou [1,3,*], Hao Li [3], Rongchuan Lv [3], Yuanchao Wu [3], Guangnan Song [3], Wenjin Wang [1] and Ren Zhai [1]

1 School of Electronic Information and Communications, Huazhong University of Science and Technology, Wuhan 430074, China; d201880571@hust.edu.cn (C.X.); m202072171@hust.edu.cn (W.W.); m201972060@hust.edu.cn (R.Z.)
2 National Satellite Ocean Application Service, Beijing 100081, China; wangxi@mail.nsoas.org.cn
3 China Academy of Space Technology (Xi'an), Xi'an 710100, China; lih19@cast504.com (H.L.); lvrc@cast504.com (R.L.); wuyc1@cast504.com (Y.W.); songgn@cast504.com (G.S.)
* Correspondence: haofeng_dou@hust.edu.cn

**Abstract:** When observing the Earth from space, the synthetic aperture radiometer antenna array is sometimes set as a non-uniform array. In non-uniform synthetic aperture radiometer image reconstruction, the existing brightness temperature image reconstruction methods include the grid method and array factor forming (AFF) method. However, when using traditional methods for imaging, errors are usually introduced or some prior information is required. In this article, we propose a new IASR imaging method with deep convolution neural network (CNN). The frequency domain information is extracted through multiple convolutional layers, global pooling layers, and fully connected layers to achieve non-uniform synthetic aperture radiometer imaging. Through extensive numerical experiments, we demonstrate the performance of the proposed imaging method. Compared to traditional imaging methods such as the grid method and AFF method, the proposed method has advantages in image quality, computational efficiency, and noise suppression.

**Keywords:** non-uniform synthetic aperture radiometer; image reconstruction; deep convolution neural network

## 1. Introduction

The application flexibility of a regular antenna array is limited because the antenna elements need to be arranged according to a certain array shape, and there will be more redundant baselines in the regular arrangement of antenna elements. In addition, when the minimum antenna spacing of a regular antenna array is greater than a half wavelength, there will be grid lobe and periodic aliasing, and its non-aliasing field of view will be narrowed [1,2].

The antenna array of a non-uniform antenna array synthetic aperture radiometer can be arranged according to needs such as according to the shape of the platform and according to the formation of satellites, so it has better flexibility and adaptability. The redundant baseline can be designed as very little or even close to zero, so the number of antennas, channels, and correlators required for the same spatial resolution can be less. Because the sampling of a non-uniform antenna array synthetic aperture radiometer in the spatial frequency domain is non-uniform, it is also called a non-uniform sampling synthetic aperture radiometer. Its array factor does not have periodicity, so its inversion image will not have obvious periodic aliasing.

However, non-uniform sampling synthetic aperture radiometer (IASR) will face the problems of a complex image reconstruction algorithm and poor inversion accuracy in application. The G-matrix reconstruction method and FFT method are often applied to a regular antenna array synthetic aperture radiometer, while a non-uniform sampling

synthetic aperture radiometer is difficult to adopt using the above method [3,4]. The reasons are as follows:

1.  The G-matrix method measures the G-matrix of the system response, and then uses the measured G-matrix to reconstruct the brightness temperature image of the measured data by using the regularization algorithm. There are many very small singular values in the G-matrix of the impulse response of a non-uniform sampling synthetic aperture radiometer system. These small singular values are caused by the non-uniform arrangement of element antennas. When the G-matrix reconstruction method is used for image reconstruction, a stable solution cannot be obtained.
2.  Based on the ideal situation, the FFT method can meet the Fourier transformation relationship between the visibility function and the brightness temperature image. Through the measured visibility function, the anti-Fourier transform is transformed and rebuilds the brightness temperature image. Although a stable solution can be obtained by using the FFT method, the FFT method requires that the sampling points in the frequency domain are evenly distributed, while the sampling points of the non-uniform sampling synthetic aperture radiometer on the UV plane are non-uniform, which will introduce large errors into the inversion image.

There are many methods for the brightness temperature image reconstruction of a non-uniform synthetic aperture radiometer including the grid method, array factor forming (AFF) method, and so on [5–10]. The grid method can improve the computational efficiency of the inversion of a non-uniform sampling synthetic aperture radiometer and obtain better inversion results. However, in terms of mathematical process, the grid method is a convolution interpolation algorithm. When using the grid method to invert the brightness temperature image, it will introduce errors including the truncation error, aliasing error, and discrete error. When using the AFF method for image reconstruction, it has certain error correction ability and will not produce an aliasing error. Compared with the grid method, the reconstruction effect is better, but there is still room for improvement. At present, convolutional neural networks can be used for both regression and classification tasks (by defining the different input, output, and loss functions of the network, the network can be used for different tasks), and there have been studies on the use of CNNs for imaging with mirror aperture synthetic imaging [11–13]. However, due to the characteristics of non-uniform synthetic aperture radiometers such as non-uniform sampling in the frequency domain and that the baseline is not necessarily an integer, the existing network structure cannot be applied to the imaging of a non-uniform synthetic aperture radiometer. It is necessary to design the corresponding convolutional neural network structure according to the characteristics of a non-uniform synthetic aperture radiometer.

Therefore, it is necessary to study the reconstruction algorithm of a non-uniform sampling synthetic aperture radiometer in order to reduce the inversion complexity and improve the inversion accuracy. Aiming at the two-dimensional non-uniform synthetic aperture radiometer system, this paper uses a convolution neural network to reconstruct the brightness temperature image of a non-uniform synthetic aperture radiometer. The proposed method was verified by simulation. When using the same dataset, the reconstruction effect was better than the traditional method, with an average RMSE of about 10.1 K and an average PSNR of about 24.4 dB (both better than the traditional grid method and AFF method). When using a small sample dataset for network training, the quality of the brightness temperature image reconstructed by the network is still acceptable. The proposed method is more suitable for the reconstruction of non-uniform synthetic aperture radiometer images.

The rest of this paper is organized as follows. In Section 2, we briefly introduce the process of non-uniform synthetic aperture image reconstruction and the definition of a related neural network. In Section 3, we introduce the steps of dataset generation, the overall network structure, and the training steps in detail. In Section 4, through different simulations, the imaging quality differences between different methods and the methods proposed in this paper are compared. The results show that the method proposed in this

paper has advantages in imaging quality and error correction. In Section 5, we summarize the content of the full text.

## 2. Related Works

### 2.1. Non-Uniform Synthetic Aperture Radiometer Model

Synthetic aperture radiometer uses a sparse antenna array to pair the unit antennas of the array to form many binary interferometers with different baselines. The binary interferometer measures the visibility function sampling through the complex correlation receiver, and finally reconstructs the original scene brightness temperature image distribution map through the inversion algorithm. The basic principle diagram of a two-dimensional distributed synthetic aperture radiometer is shown in Figure 1 [14–16].

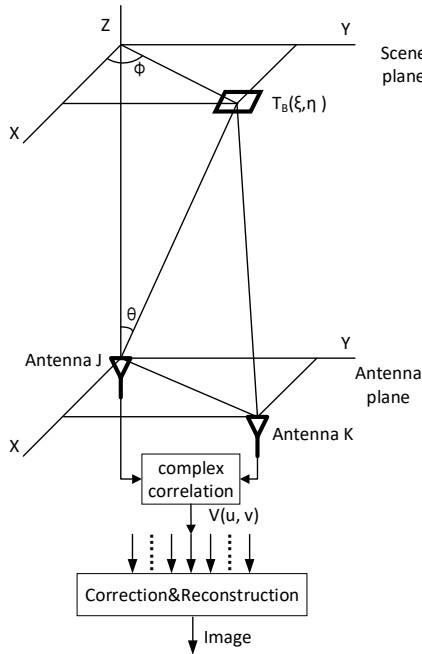

**Figure 1.** Schematic of the interference measurement.

In IASR, the measurement output is called the visibility function, given as Equation (1):

$$V(u,v) = \iint\limits_{\xi^2+\eta^2\leq1} T(\xi,\eta)e^{-j2\pi(u\xi+v\eta)}d\xi d\eta, \tag{1}$$

where $T(\xi,\eta)$ is the modified brightness temperature distribution of the observed scene; $(\xi,\eta)$ are the coordinates of the brightness temperature distribution; the obtained visibility function $V(u,v)$ is a complex-valued matrix, $(u,v)$ are the spatial frequency determined by the spacing between the antenna positions; $(\xi,\eta) = (\sin\theta\cos\varphi,\ \sin\theta\sin\varphi)$ is the direction cosine; $\theta$ and $\varphi$ are the azimuth. An estimate of the brightness temperature distribution can be reconstructed by inverse discrete Fourier transform:

$$T(\xi,\eta) = \sum_{i=-I}^{I}\sum_{j=-J}^{J} V(u_i,v_j)e^{j2\pi(u_i\xi+v_j\eta)}, \tag{2}$$

### 2.2. Definition of the Related Neural Network

(1) Convolution

In a typical convolution process, the input data are convolved in parallel with multiple learnable filters (convolutional kernels). The result of the convolution is sent to a nonlinear

activation function such as a sigmoid or a rectified linear unit (ReLU) to generate as many feature maps as the convolution filter. Hidden units in the same feature map share the same weights and can be displayed as feature extractors that detect specific features at each location in the previous layer. Therefore, each feature map represents unique features at different locations in the previous layer. The details of the complex number convolution process can be displayed as follows [17–20].

In the convolution layer, the input feature maps $O_i^{l-1}(i = 1, \ldots, I)$ of the previous layer are connected to the output feature maps $O_j^l(j = 1, \ldots, J)$ of the current layer, and $O_j^l$ can be calculated by:

$$O_j^l = BN(\sum_{i=1}^{I} w_{ji}^l * O_i^{l-1} + b_j^l), j = 1, \ldots, J \ \ i = 1, \ldots, I \tag{3}$$

where $BN$ denotes the nonlinear activation function; $*$ represents the convolutional operation; $w_{ji}^l$ represents the bank of filters (convolutional kernel); $b_j^l$ represents the bias of the convolutional kernels in layer $j$; $i$ and $j$ represent the serial number of the input and output channels of the $l$-th convolution layer; $I$ and $J$ represent the total number of the input and output channels of the $l$-th convolution layer.

The hyperparameters of the convolutional layer include the number of feature maps (J), filter size (F), stride (S), and zero-padding (P). The filter size (F) is the size of the convolutional kernel. The stride (S) is the distance that the convolution kernel moves each time on the input image. Zero-padding (P) means adding an appropriate number of zeros to the edges in the input image. When the size of the input image of layer $j$ is $X_1 \times Y_1$, the size of the output image of layer $j$ is $X_2 \times Y_2$, where $Y_2 = (Y_1 - F + 2P)/S + 1$ and $X_2 = (X_1 - F + 2P)/S + 1$.

(2) Global average pooling

The features extracted by the convolution operation are local features (the nodes of the convolution layer are only connected to some nodes in the previous layer), and the receptive field is local, so the convolution operation cannot make good use of some information outside the receptive field [21,22].

To alleviate this problem, on each channel, global average pooling is performed on the extracted features. For each element of $Z_J$ it is represented as follows:

$$Z_J = \frac{1}{W \times H} \sum_{x=1}^{W} \sum_{y=1}^{H} O_J(x, y), \tag{4}$$

where $Z_J$ is the result of performing the global average pooling on feature $O_J$ in the spatial dimension $W \times H$.

(3) Feature enhancement

In order to utilize the information extracted by the global pooling operation and exploit the information dependencies between channels, the feature enhancement operation is used to obtain the importance of each feature channel, and the operation of the feature enhancement can be defined as follows.

$$S_J = Sigmoid(W_2 Relu(W_1 Z_J + B_1) + B_2), \tag{5}$$

where $W_1$ and $B_1$ denote the weight matrix and the bias vector of the full connection layer 1, respectively; $W_2$ and $B_2$ denote the weight matrix and the bias vector of full connection layer 2, respectively; $Relu$ and $Sigmoid$ are the nonlinear activation function.

## 3. Dataset and Learning for ISAR-CNN

In this section, we first introduce the generation steps of the simulated IASR image dataset, and then we introduce the specific structure and learning details of the IASR-CNN network.

### 3.1. Dataset Generation

The ideal IASR system simulation program (without system errors) is used to generate the dataset, which was used in the simulations. In this paper, we chose a randomly distributed non-uniform antenna array composed of 51 antennas, and the maximum baseline was 50 (after wavelength normalization). When the number of antennas is too small, the image reconstruction quality is poor due to the few sampling frequency points, and when the number of antennas is too large, the simulation calculation time is too long. In order to avoid excessive calculation and obtain a better image display, we chose a randomly distributed 51-element antenna array in our manuscript. The generating steps for the training dataset and the testing dataset are as follows.

1.  From the UC Merced (UCM) dataset (created by researchers at the University of California, Merced, CA, USA) of 21 different categories of remote sensing images, an image was selected and converted to grayscale. In order to correspond to the scale of the conventional microwave brightness temperature image, we mapped the scale of the image from 0–255 to 2.7–300 (in the practical application of Earth remote sensing, the scene brightness temperature is mostly distributed between 2.7 K and 300 K, and the simulated scene covers the dynamic range of the whole scene brightness temperature distribution). Then, we used this image as the original scene brightness temperature image ($T$).
2.  $T$ is then input into the ideal non-uniform synthetic aperture radiometer simulation program to generate the simulated non-uniform synthetic aperture radiometer visibilities function $V(u,v)$ and its corresponding UV plane coordinates $(u,v)$. In the non-uniform synthetic aperture radiometer simulation program, the antenna array is set to a randomly distributed 51-element antenna array, and the antenna array is shown in Figure 2 (the X-axes and Y-axes represent the relative position of the antenna distribution, and the unit is the multiple of the wavelength). The receiving frequency was set to 33.5 GHz.

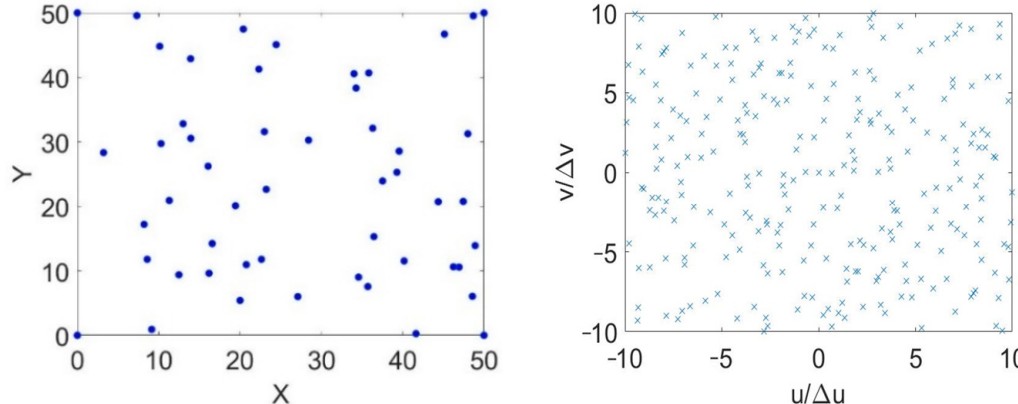

**Figure 2.** Randomly generated antenna array (after wavelength normalization) and UV plane sampling point.

3.  $V(u,v)$ (size of $1 \times 269$) and $(u,v)$ (size of $1 \times 269$) were combined as one input data $[V(u,v) \, (u,v)]$, with a size of $3 \times 269$.
4.  $[V(u,v) \, (u,v)]$ and $T$ (size of $79 \times 79$) were combined as one sample.
5.  After each image in the UCM dataset undergoes steps 1–4, we can obtain multiple samples to form dataset (D).
6.  Randomly select 90% of the data from dataset (D) as the training dataset ($D_{train}$), and the remaining 10% of the data from dataset (D) is then used for network testing as the testing dataset ($D_{test}$)

### 3.2. Learning for IASR-CNN

Aiming at addressing the problem of two-dimensional non-uniform synthetic aperture radiometer brightness temperature image reconstruction, the input and output data of the convolutional neural network were specifically defined. The input data were defined as the spatial frequency data (the visibility function $V(u,v)$) and the corresponding spatial frequency coordinates (the coordinates $(u,v)$ of the UV plane), and the output data were the microwave brightness temperature image ($T$).

By training the network with a suitable training dataset, the network can learn the mapping relationship between the frequency and the spatial, realize the transformation from the frequency to the spatial, and reconstruct the microwave brightness temperature image, which is called the IASR-CNN. The IASR-CNN consists of two parts: the information extraction (consist of 1D-CNN, 2D-CNN, global average pooling, and feature enhancement) and the image reconstruction (consist of 2D-CNN). The structure of the IASR-CNN is depicted in Figure 3.

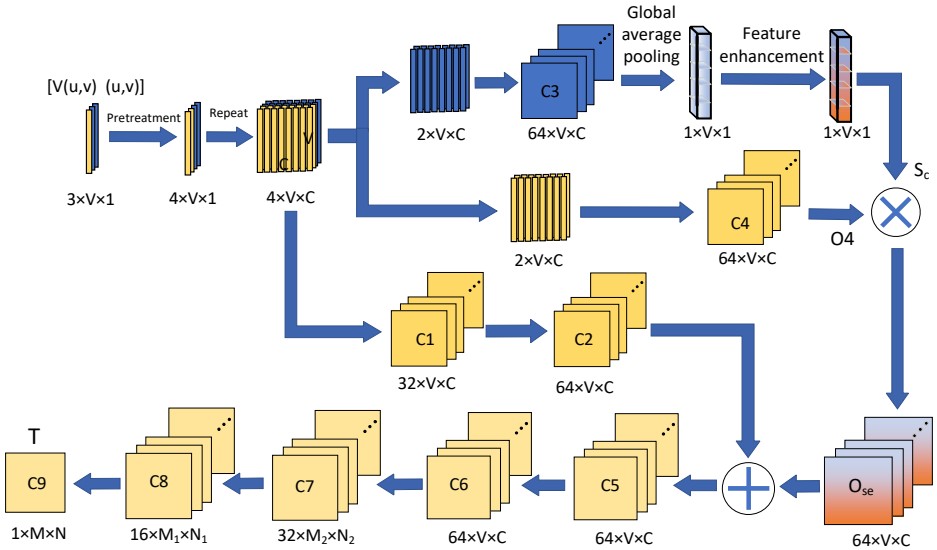

**Figure 3.** Structure of the IASR–CNN.

Because the function of the network is to reconstruct the microwave brightness temperature image rather than classification, the pool layer and full connection layer are not used in the image reconstruction stage of the network. The brightness temperature image is obtained directly after dimensionality reduction through a multi-layer convolution layer [23]. The following describes the network in detail.

The input data are combined by $V(u,v)$ and $(u,v)$, which have sizes of $1 \times 269$ and $2 \times 269$. Since the visibility function $V(u,v)$ is a complex matrix, and the convolutional neural network cannot process the complex matrix, the visibility function has to be reorganized into a real-valued matrix with a size of $2 \times 269$. After pretreatment, the size of the input data changes from $3 \times 269$ to $4 \times 269$, which is composed of the real part of the visibility function, the imaginary part of the visibility function, the $u$ coordinate, and $v$ coordinate, respectively. Then, the preprocessed data (size of $4 \times 269$) are repeated to a size of $4 \times 269 \times 269$. After the repeat, the input data have undergone two convolution operations. Convolution layer 1 (C1) contains 32 filters, and the size of each filter is $3 \times 3$. Convolution layer 2 (C2) contains 64 filters, and the size of each filter is $3 \times 3$. In convolution layers 1 and 2, the stride of the filters is 1 and the pooling size is $1 \times 1$.

At the same time, the repeated data (size of $4 \times 269 \times 269$) are divided into the visibility function part (size of $2 \times 269 \times 269$) and UV plane coordinate part (size of $2 \times 269 \times 269$). The UV plane coordinate part is input to the convolutional layer (C3). C3 contains 64 filters, each with a size of $3 \times 3$. The stride of the filters is 1, and the pooling size is $1 \times 1$. Then, after global average pooling and feature enhancement of the output of C3, the channel

weighting information ($S_c$) is obtained. At the same time, the visibility function part (size of $2 \times 269 \times 269$) is input to the convolutional layer (C4), and the output is the convolution features between channels ($O_4$). The convolutional layer (C4) contains 64 filters, each with a size of $3 \times 3$. The stride of the filters is 1 and the pooling size is 0.

In order to make better use of the global information for the features between channels, $O_4$ and $S_c$ are multiplied to obtain the weighted channel information ($O_{se}$) for subsequent processing.

The outputs of $O_{se}$ and C2 are added and input into the convolution layer (C5). The convolutional layers (C5, C6, C7, C8, and C9) are connected as shown in Figure 3. The convolutional layer (C5) contains 64 filters, each with a size of $5 \times 5$. The stride of the filters is 1 and the pooling size is $2 \times 2$. The convolutional layer (C6) contains 64 filters, each with a size of $3 \times 3$. The stride of the filters is 1 and the pooling size is $2 \times 2$. The convolutional layer (C7) contains 32 filters, each with a size of $5 \times 5$. The stride of the filters is 2 and the pooling size is $1 \times 1$. The convolutional layer (C8) contains 16 filters, each with a size of $5 \times 5$. The stride of the filters is 2 and the pooling size is $1 \times 1$. The convolutional layer (C9) contains one filter, each with a size of $3 \times 3$. The stride of the filters is 1 and the pooling size is $1 \times 1$.

Batch normalization (BN) and rectified linear unit (ReLU) operation are included after the convolution operation of convolution layers C1–C8, and the Sigmoid operation is included after the convolution operation of convolution layer C9. ReLU and Sigmoid are activation functions, whose formulas are ReLU $(m)$ = Max $(0, m)$ and Sigmoid $(m) = 1/(1 + e^{-m})$, respectively.

After the IASR-CNN structure is determined, the network needs to be trained through the training dataset generated in Section 3.1 to determine the parameters (such as weights and biases) of the network. The training dataset $D_{train}$ contains multiple samples. Each sample is composed of a non-uniform synthetic aperture radiometer visibilities function $V(u, v)$ as well as its corresponding UV plane coordinates $(u, v)$ and corresponding original scene brightness temperature image ($T$). The generation of the dataset is described in detail in the previous section.

The input of the IASR-CNN is the visibilities function $V(u, v)$ and its corresponding UV plane coordinates $(u, v)$. The output of the IASR-CNN is the brightness temperature image ($T_{cnn}$). When all the samples in the training dataset $D_{train}$ are input into the network, the network completes an epoch training. Through multiple epoch training, the parameters in the IASR-CNN are adjusted iteratively to reduce the root mean square error (RMSE) between the $T_{cnn}$ and $T$ (in the training dataset $D_{train}$). When the number of training cycles reaches the set epoch or the RMSE tends to be stable, it indicates that the network has completed the training. In this paper, by comparing the final convergence results of different gradient descent algorithms (such as the SGD, Adadelta, and Adam algorithms), the standard backpropagation stochastic gradient descent algorithm (SGD) was selected to minimize the RMSE and update the IASR-CNN parameters (the learning rate attenuation strategy will be adopted in the actual use, so the learning rate decreases gradually). The formula for calculating the RMSE between $T_{cnn}$ and $T$ (in the training dataset) is as follows.

$$RMSE = \frac{1}{L} \sum_{k=1}^{L} \sqrt{\frac{1}{X \times Y} \sum (T_{cnnk} - T_k)^2} \tag{6}$$

where the size of the $T$ and $T_{cnn}$ is $X \times Y$; $k$ is the sample sequence number; $L$ is the total number of samples in the training dataset.

## 4. Experiments and Result Analysis

In this section, we used the IASR-CNN for non-uniform synthetic aperture radiometer imaging and compared it with traditional imaging methods (the grid method and AFF method).

The IASR-CNN was programmed under the Pytorch framework based on Python 3.8, and CUDA11.1 (Compute Unified Device Architecture) was used to accelerate the GPU.

The configuration of the computer hardware was AMD Ryzen 9 5950X, with 64 GB RAM and a GPU card (NVIDA GeForce RTX 3090).

### 4.1. Ideal Simulation Results

Through the error-free dataset generated by simulation, the network was trained and tested to verify whether the IASR-CNN could learn the mapping relationship between the visibilities function $V(u, v)$ and its corresponding UV plane coordinates $(u, v)$ and the original scene brightness temperature ($T$) as well as reconstruct the brightness temperature image. The effect of the network reconstruction of brightness temperature image was evaluated qualitatively and quantitatively.

The training dataset generated in Section 3.1 was used to train the IASR-CNN of the 51-element non-uniform array. Figure 4 shows the RMSE convergence trend in the IASR-CNN training process (it took about 7 h to complete the training). Obviously, the network was relatively stable after 300 iterations, and the convergence speed of the IASR-CNN was fast. The RMSE converged to about 7.4 K.

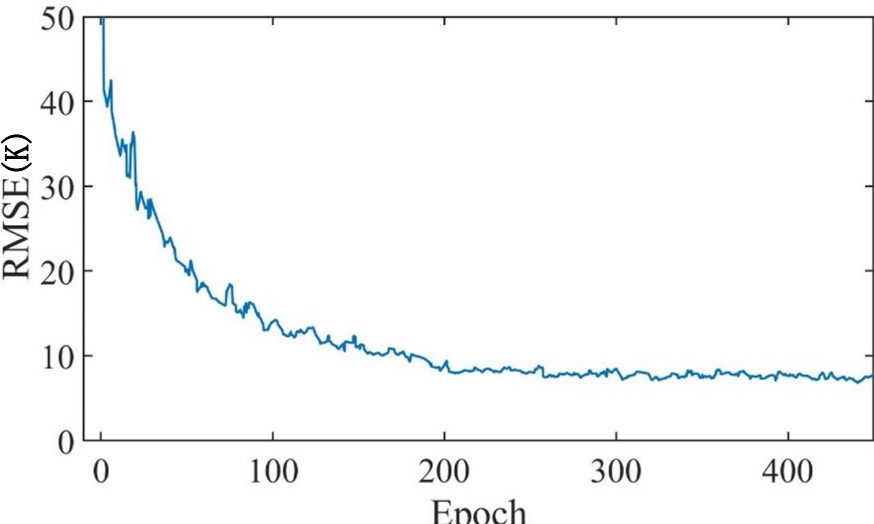

**Figure 4.** The RMSE convergence trend.

We used the test dataset (generated in Section 3.1) to test the trained network to verify that the network could reconstruct the brightness temperature. Figure 5a shows the original scene brightness temperature ($T$), which was randomly selected from the test datasets, Figure 5b shows the image $T_{cnn}$ reconstructed by the network, Figure 5c shows the image $T_{\text{grid}}$ reconstructed by the grid method, and Figure 5d is the image $T_{\text{aff}}$ reconstructed by the array factor forming method. A comparison with the original scene brightness temperature showed that the effect of the IASR–CNN method for the brightness temperature reconstruction was better, and visually, the $T_{cnn}$ reconstructed by the network was not much different from the original scene image $T$.

An objective index for the evaluation, root mean square error (RMSE), and peak signal-to-noise ratio (PSNR) is often used to analyze the quality of the brightness temperature image reconstructed by different methods. When the RMSE is smaller, the smaller the error between the original scene brightness temperature image and the image reconstructed by different methods. The RMSE is usually used to quantitatively reflect the brightness temperature reconstruction effect of different methods. The PSNR is usually used to measure the distortion degree of the image reconstructed by different methods. The larger the PSNR, the smaller the distortion of the image reconstructed by different methods, and the formula is as follows.

$$PSNR = 10\log_{10}\left(\frac{D^2 \times M \times N}{\sum(T_{cnn} - T)^2}\right) \tag{7}$$

The RMSE of the IASR-CNN method, grid method, and AFF method were 10.1625 K, 15.3871 K, and 14.9078 K, respectively, and the PSNR values were 24.41 dB, 20.57 dB, and 21.66 dB, respectively. It can be seen that the RMSE value of the network method was less than that of the grid method and AFF method, while the PSNR value was higher than that of the grid method and AFF method. Therefore, the quality of the image reconstructed by the IASR–CNN method was better than that of the two traditional methods (the grid method and the AFF method).

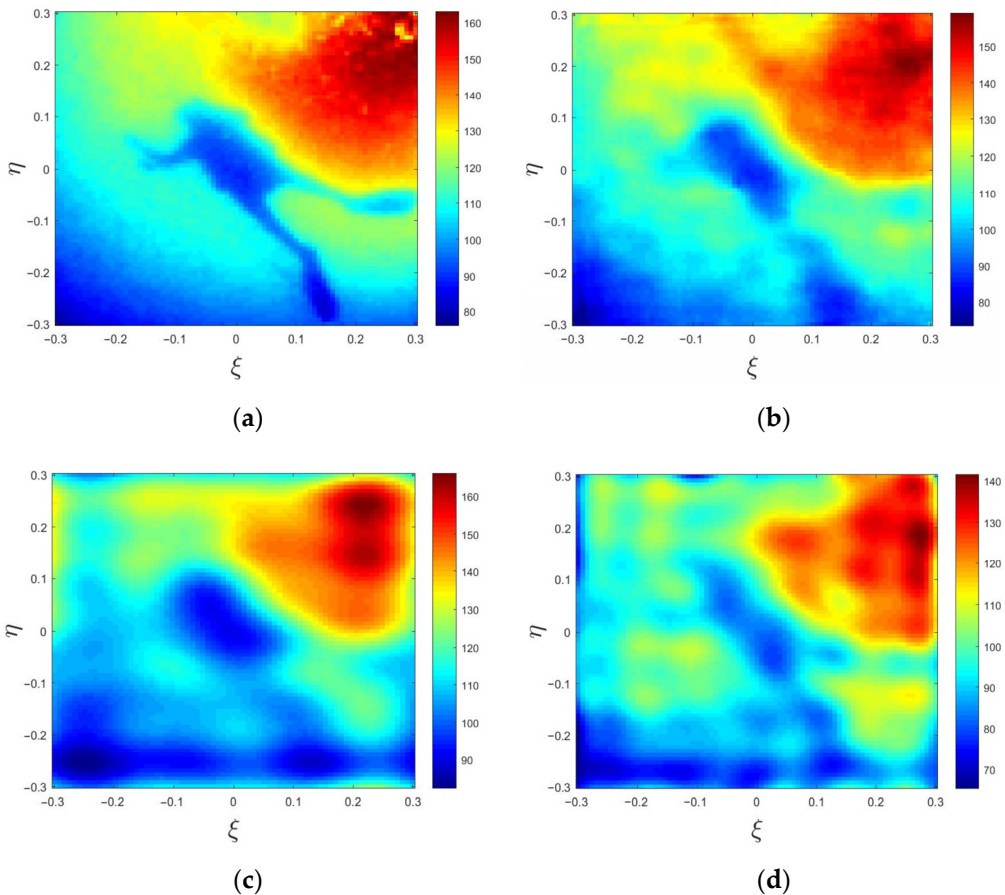

**Figure 5.** The IASR images reconstructed by different methods at the same scene. (**a**) Original scene. (**b**) IASR-CNN method. (**c**) Grid method. (**d**) AFF method.

In order to further analyze the spectrum of the reconstructed image by the three methods, after normalizing the spectrograms of the images in Figure 5a–d, one-dimensional curves were intercepted along the u and v directions through the maximum value points (the Y-axes represent the normalized spectral amplitude), as shown in Figure 6a,b. Since the spectral differences in Figure 6a,b were not obvious, a partial enlarged view of the high-frequency region in Figure 6a,b was drawn, as shown in Figure 6c,d.

It can be seen from Figure 6a–d that there was a clear gap between the spectrum of the image reconstructed by the grid or AFF method and the spectrum of the original image, regardless of whether it was in the $u$ direction or the $v$ direction, and the spectrum of the image reconstructed by the IASR-CNN was closer to the spectrum of the original scene. The training label of the IASR-CNN network is the original scene brightness temperature image, and the image contained all of the spectral information. Through the network structure such as information extraction and inter-channel feature enhancement, the network learns as much of the spectrum information as possible from the original scene brightness temperature image, and corrects the truncation errors due to non-uniform synthetic aperture radiometer measurements.

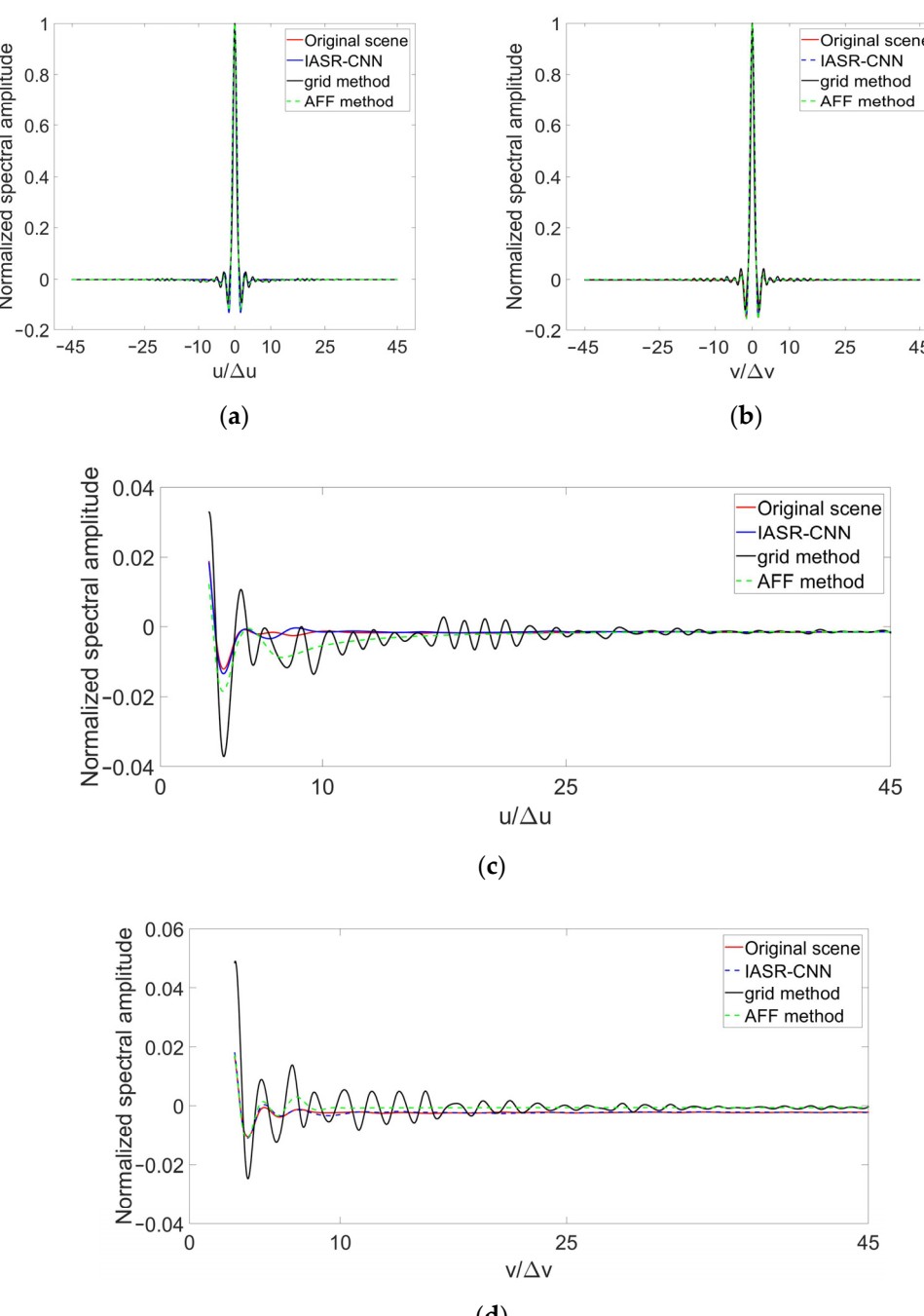

**Figure 6.** The spectrum of images reconstructed by different methods at the same scene. (**a**) Spectrum in direction *u*. (**b**) Spectrum in direction *v*. (**c**) High-frequency spectrum in direction *u*. (**d**) High-frequency spectrum in direction *v*.

We used the RMSE to quantitatively compare the spectrum of the image reconstructed by different methods (the IASR-CNN method, grid method, and AFF method) with the original scene spectrum, and the RMSEs were $1.31 \times 10^{-3}$, $2.15 \times 10^{-4}$ and $1.77 \times 10^{-4}$ (after spectrum normalization). This quantitatively shows that the spectrum of the network reconstructed image was closer to the original scene spectrum.

The above evaluation metrics were only calculated based on a single sample in the test dataset generated in Section 3.1, which was not enough to explain the overall situation of the test dataset and the effect of the network reconstruction. Therefore, 1000 samples of data were selected from the test dataset, and we used different methods (the IASR-CNN,

grid, and AFF methods) to reconstruct the IASR brightness temperature image from the visibility function in the sample, respectively.

The reconstruction results of 1000 samples were counted, and the average RMSE and PSNR of the different methods were calculated, respectively. The results are shown in Tables 1 and 2.

**Table 1.** The evaluation metrics of the brightness temperature images reconstructed by three methods.

| Method | RMSE (K) | PSNR (dB) |
|---|---|---|
| IASR–CNN method | 9.4718 | 24.6375 |
| Grid method | 15.3268 | 20.6268 |
| AFF method | 14.8785 | 21.8099 |

**Table 2.** The evaluation metrics of the spectrum corresponding to the images reconstructed by three methods.

| Method | RMSE |
|---|---|
| IASR–CNN method | $1.35 \times 10^{-4}$ |
| Grid method | $2.21 \times 10^{-4}$ |
| AFF method | $1.76 \times 10^{-4}$ |

As can be seen from Table 1, the average RMSE value of the IASR-CNN method was lower than the two traditional methods, while the average PSNR value was higher than the two traditional methods. Therefore, the quality of the brightness temperature image reconstructed by the IASR-CNN method was higher than that of the grid method and AFF method.

In the case of ignoring the measurement error and noise interference, due to the lack of spatial frequency measurement in the two-dimensional non-uniform synthetic aperture radiometer system, the grid method will interpolate the visibility function and introduce additional errors into the image reconstruction. The AFF method cannot supplement and correct the lack of spatial frequency measurement, which leads to the imaging quality of these two methods not being as good as that of the IASR-CNN method.

As can be seen from Table 2, when reconstructing the brightness temperature image, the IASR-CNN method can not only learn the mapping relationship between the non-uniform visibility function and the corresponding UV plane coordinates and the brightness temperature image of the original scene from the training dataset, but also supplements and corrects the missing frequency domain components measured by the non-uniform synthetic aperture radiometer system. The spectrum of the brightness temperature image reconstructed by the IASR-CNN method was closer to the spectrum of the original scene compared to the spectrum of the brightness temperature image reconstructed by the grid method or the AFF method.

Although it takes a long time to train the network, when the network is trained, the time required to reconstruct the brightness temperature image using the trained network is shorter, which is faster than the traditional method, as shown in Table 3.

**Table 3.** Time required for image reconstruction by the different methods.

| Method | Time (s) |
|---|---|
| IASR–CNN method | 0.016 |
| Grid method | 0.19 |
| AFF method | 0.85 |

*4.2. Simulation Results with Errors*

In this section, we added Gaussian noise to the ideal visibility function generated by the simulation, and then used the trained network to reconstruct the brightness temperature

images. Through a comparison with the grid method and the AFF method, we verified the noise immunity and correction of the network.

Gaussian noise with an intensity of 0 to 0.3 (i.e., a mean of 0 and a standard deviation of 0 to 0.3) was added to the visibility function of the sample. One sample was randomly selected from all of the samples in the test dataset. Under different noise intensity, three methods (the IASR-CNN method, grid method, and AFF method) were used to reconstruct the microwave brightness temperature image of the input data (including visibility function and corresponding UV plane coordinates) in the sample. The results are shown in Figure 7. Under different noise intensities after imaging, all of the data in the test dataset by the three methods, the average RMSE, and PSNR are shown in Figure 8.

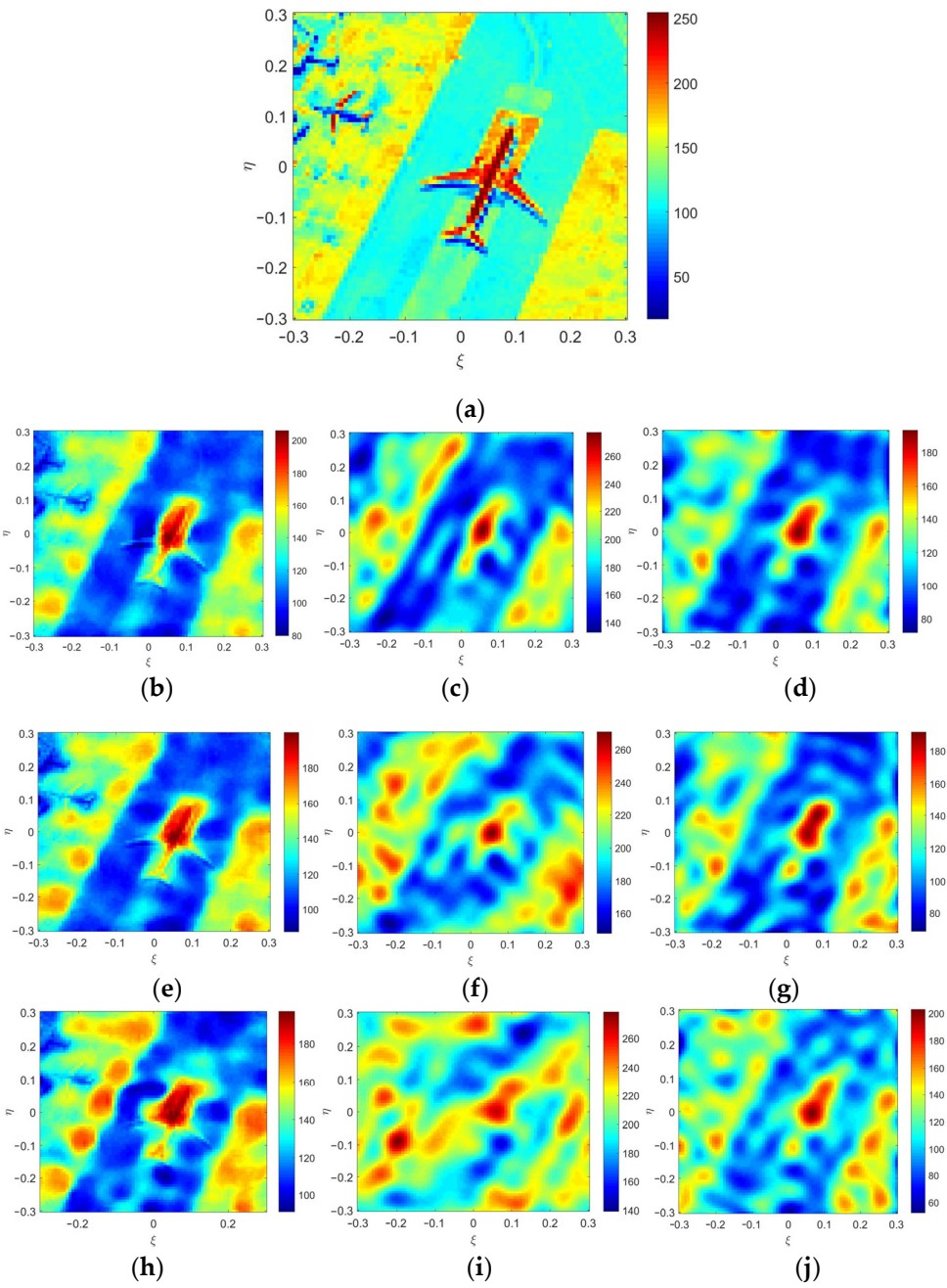

**Figure 7.** The brightness temperature images reconstructed by the different methods at variable noise intensities. (**a**) Original scene. (**b**) The IASR-CNN method at a 0.1 noise intensity. (**c**) The grid method

at a 0.1 noise intensity. (**d**) The AFF method at a 0.1 noise intensity. (**e**) The IASR-CNN method at a 0.2 noise intensity. (**f**) The grid method at a 0.2 noise intensity. (**g**) The AFF method at a 0.2 noise intensity. (**h**) The IASR-CNN method at a 0.3 noise intensity. (**i**) The grid method at a 0.3 noise intensity. (**j**) The AFF method at a 0.3 noise intensity.

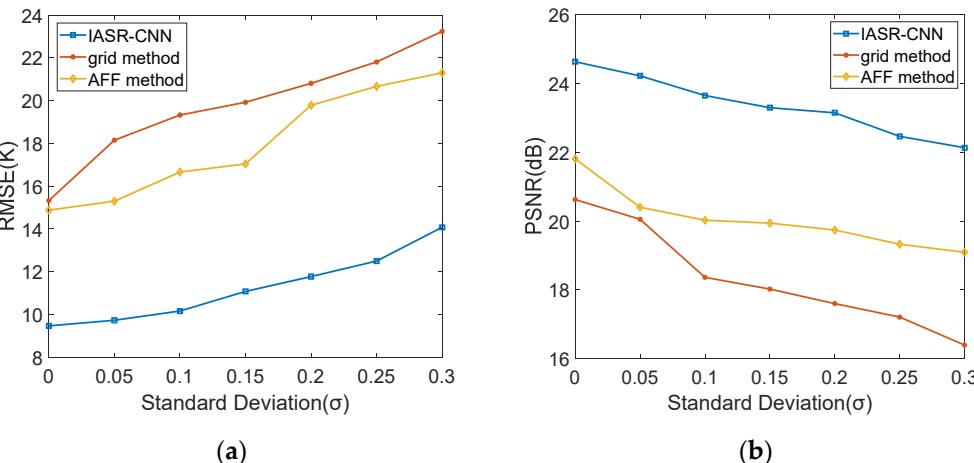

**Figure 8.** The evaluation metrics of the brightness temperature image reconstructed by different methods at variable noise intensities. (**a**) PSNR. (**b**) RMSE.

Figures 7 and 8 show that the IASR-CNN had better noise immunity than the other methods. First, for all of the noise intensities, it can be seen that the IASR-CNN always obtained a higher PSNR and lower RMSE than the other methods (the grid method and AFF method).

In addition, it should be noted that when the noise intensity increased, the RMSE of the other two methods of reconstructing the brightness temperature image increased rapidly and the PSNR decreased rapidly. The RMSE of the brightness temperature image reconstructed by the IASR-CNN was always lower than 14 K and the PSNR was always higher than 22 dB, which clearly showed its strong robustness to noise. When the noise intensity was 0.3, the gap between the IASR–CNN and the other two traditional methods was about 35% in the RMSE and 23% in the PSNR.

Next, we evaluated the performance of the network by training the network using training datasets with different usage rates (contained 50% to 90% of the samples in the full training dataset). The network was trained and tested under different noise conditions (noise free and the noise intensities from 0 to 0.3), and the results are shown in Tables 4 and 5.

**Table 4.** The RMSE (K) of the IASR-CNN method with different usage ratios.

| Variable Noise Intensities | 50% | 60% | 70% | 80% | 90% |
| --- | --- | --- | --- | --- | --- |
| 0 | 9.8924 | 9.7922 | 9.7757 | 9.6031 | 9.5327 |
| 0.1 | 10.7056 | 10.6691 | 10.5176 | 10.3349 | 10.2615 |
| 0.2 | 12.8341 | 12.7241 | 12.2785 | 12.0975 | 11.9740 |
| 0.3 | 15.3358 | 15.1270 | 14.9058 | 14.5469 | 14.1078 |

**Table 5.** The PSNR (dB) of the IASR-CNN method with different usage ratios.

| Variable Noise Intensities | 50% | 60% | 70% | 80% | 90% |
| --- | --- | --- | --- | --- | --- |
| 0 | 23.9786 | 24.0357 | 24.2759 | 24.3040 | 24.3436 |
| 0.1 | 23.1031 | 23.2295 | 23.3485 | 23.4191 | 23.5625 |
| 0.2 | 22.0849 | 22.5657 | 22.6787 | 22.8003 | 22.9299 |
| 0.3 | 20.3953 | 20.8124 | 21.1419 | 21.3134 | 22.0037 |

As can be seen from Tables 4 and 5, the reconstruction quality of the brightness temperature image with the non-uniform synthetic aperture radiometer was still acceptable when the proposed method used a small sample of the training dataset.

We found that when the complete training dataset was used to train the network, the performance of the IASR-CNN was much better than that of traditional imaging methods, which showed that the IASR-CNN could learn more useful frequency domain information from the samples of the training dataset (including visibility function, corresponding UV plane coordinates, and the original scene brightness temperature), and the traditional method will be affected by the information loss and artifacts caused by the non-uniform synthetic aperture radiometer inverse imaging process, resulting in poor imaging quality.

On the other hand, when we used 50% of the training samples to train the network, and the noise intensity was 0.1, the brightness temperature image reconstruction quality of the IASR-CNN was 26% and 15% (PSNR) higher and 44% and 35% (RMSE) lower than the grid method and the AFF method, respectively. When the noise intensity was 0.2, the brightness temperature image reconstruction quality of the IASR-CNN was 24% and 11% (PSNR) higher and 38% and 34% (RMSE) lower than the other two traditional methods, respectively. When the noise intensity was 0.3, the brightness temperature image reconstruction quality of the IASR-CNN was 23% and 7% (PSNR) higher and 33% and 27% (RMSE) lower than the other two traditional methods, respectively.

## 5. Conclusions

The array factor of the non-uniform sampling synthetic aperture radiometer does not have periodicity, so its inversion image will not have obvious periodic aliasing. However, the non-uniform sampling synthetic aperture radiometer will face the problems of the complex image reconstruction algorithm and poor inversion accuracy in application, and the G-matrix reconstruction method and the FFT method that are often applied to a regular antenna array synthetic aperture radiometer are difficult to apply to a non-uniform antenna array synthetic aperture radiometer.

This paper proposed a deep convolutional neural network method for IASR image reconstruction, called the IASR-CNN method. The network consists of multiple fully connected layers, multiple convolutional layers, and a global average pooling layer. This method uses a specially designed convolutional neural network to learn the mapping relationship between the visibility function obtained by the IASR and the original scene and the systematic error during observation, in order to improve the imaging quality of the IASR images.

Ideally, using an error-free dataset to train and test the network, the IASR-CNN method reconstructs images of better quality than traditional imaging methods (the grid method and AFF method). The reconstructed image quality of the different methods was quantitatively compared. A total of 1000 samples in the test dataset were used to test the three methods (the IASR-CNN method, the grid method, and the AFF method), and the average RMSE was 9.4718 K, 15.3268 K, and 14.8785 K, respectively, and the average PSNR was 24.6375 dB, 20.6268 dB, and 21.8099 dB, respectively. After the imaging, the image was transferred to the frequency domain, the RMSE of all samples in the test dataset was counted, and the average RMSE was $1.35 \times 10^{-4}$, $2.21 \times 10^{-4}$ and $1.76 \times 10^{-4}$, which quantitatively showed that the image spectrum reconstructed by the IASR-CNN method was closer to the original scene spectrum compared to the traditional method (the grid method and the AFF method). The quantitative results showed that the reconstructed image quality of the IASR-CNN method was the best.

The simulation results with the errors showed that the brightness temperature image reconstruction quality (50% of the training samples are used) of the IASR-CNN was 26% and 15% (PSNR) higher and 44% and 35% (RMSE) lower than the grid method and the AFF method, respectively, when the noise intensity was 0.1. When the noise intensity was 0.2, the brightness temperature image reconstruction quality of the IASR-CNN was 24% and 11% (PSNR) higher and 38% and 34% (RMSE) lower than the other two traditional

methods, respectively. When the noise intensity was 0.3, the brightness temperature image reconstruction quality of the IASR-CNN was 23% and 7% (PSNR) higher and 33% and 27% (RMSE) lower than the other two traditional methods, respectively.

When using a small sample dataset for the IASR-CNN network training, the quality of the brightness temperature image reconstructed by the IASR-CNN method was acceptable. Compared to the grid method and the AFF method, the IASR-CNN method is more suitable for the reconstruction of a non-uniform synthetic aperture radiometer brightness temperature image.

**Author Contributions:** Conceptualization, C.X. and H.D.; methodology, C.X.; software, C.X.; validation, C.X., H.D., W.W. and R.Z.; formal analysis, C.X.; investigation, C.X.; resources, C.X.; data curation, C.X.; writing—original draft preparation, C.X.; writing—review and editing, C.X. and H.D.; visualization, C.X.; supervision, C.X., H.D., X.W., H.L., R.L., Y.W., G.S., W.W. and R.Z.; project administration, C.X.; funding acquisition, H.D. All authors have read and agreed to the published version of the manuscript.

**Funding:** This work was supported in part by the Qian Xuesen Youth Innovation Fund (QXSQNCXJJ2020-504), and in part by the China Postdoctoral Science Foundation (2021M703491). The APC was funded by China Academy of Space Technology (Xi'an).

**Conflicts of Interest:** The authors declare no conflict of interest.

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
