# Peer review of "Non-Uniform Synthetic Aperture Radiometer Image Reconstruction Based on Deep Convolutional Neural Network"

_remotesensing, doi:10.3390/rs14102359_

Round 1

Reviewer 1 Report

The authors have done a great job and the new version of the manuscript is significantly improved compared to the original. I am satisfied with the responses to my comments and can recommend the work for publication.

Reviewer 2 Report

The authors seem to have addressed all comments from the previous review rounds and thus this revised manuscript may be accepted for publication as-is.

Reviewer 3 Report

Regarding the revisions made by the authors, the previous concerns are addressed so I suggest acceptance of the paper in the current form.

Reviewer 4 Report

The authors made an effort to satisfactorily improve the paper in accordance with the recommendations of the reviewers. Therefore, the paper can be recommended for publication

This manuscript is a resubmission of an earlier submission. The following is a list of the peer review reports and author responses from that submission.

Round 1

Reviewer 1 Report

In this paper, the authors propose a deep convolutional neural network method for IASR image reconstruction, called the IASR-CNN method. Simulation results, presented by the authors, show the advantages of using the proposed method compared to traditional imaging methods (grid method and AFF method). The paper contains a consistent theoretical background, useful results, as well as satisfactory scientific contributions that qualify it for publication in a journal. The following comments are addressed to the authors:

  • Graphics, tables and text lack the formal unit designation for RMSE and PSNR variables.
  • The authors state:“ We use the standard backpropagation stochastic gradient descent algorithm (SGD) to minimize RMSE and update IASR-CNN parameters“. The authors should explain why they chose the standard gradient descent as the training algorithm and not some of the algorithms from the optimized gradient descent group, such as the Adam optimizer, which has generally proven to be a better choice in solving similar problems.

Reviewer 2 Report

The authors proposed a CNN-based IASR imaging method to improve image quality, computational efficiency, and noise suppression. There are a few items to improve the quality of the paper:

1)Lines 43-47: split such long sentences into several smaller ones and for this special case, finish the sentence with ":" that shows you want to introduce some items. 

2)Lines 75-80: in this paragraph, i.e., the last paragraph of the Introduction, it is highly recommended to express clearly the paper's contributions and novelties followed by a brief summary of the paper's organization and structure.

3)Using CNNs, it is recommended to highlight the fundamental differences between classification and regression tasks. For example, you can extract some useful information from the following paragraph from "A Survey on the Applications of Convolutional Neural Networks for Synthetic Aperture Radar: Recent Advances"
"CNNs can be used for both regression and classification tasks. These two tasks are subsets of supervised ML. The main goal of a regression task is to predict continuous (numerical) variables, whereas classification is used for the prediction of discrete (categorical) variables. The main difference is the choice of the loss functions. In regression problems, mean squared error (MSE), root mean squared error (L2), mean absolute error (L1) and Huber loss are usually used, whereas, in classification problems cross-entropy loss and Hinge loss are usually employed." 

4) Lines 438-443: So many details and also the tenses in this paragraph is more suitable for the Introduction than the Conclusion.

5) Why the method has not been tested with real data?

Reviewer 3 Report

The authors propose deep learning based reconstruction of non-uniform synthetic aperture radar data, and they demonstrate the effectiveness and noise-robustness of their method using simulated data and distortions. This reviewer feels the following points need to be considered by the authors while preparing the revised version of their manuscript:

  1. Revisiting their literature survey part and look at more recent related works like https://doi.org/10.1117/1.JRS.15.026511
  2. Since the experiments are on synthetic data and synthetic perturbations, it would be worthwhile to add some comments in the results or discussion section, on how well the authors think the simulations are able to cover all the real-world scenarios and corner cases encountered by practitioners.
  3. It would be interesting as future work for other researchers if the authors add a discussion (in their revised manuscript) on how learning-based filtering or signal metrics like https://doi.org/10.1109/ICSENS.2018.8589920 could be adapted to the specific data type and problem domain of the manuscript under review, to further improve it in future.

Reviewer 4 Report

In the paper, the authors propose a new method of non-uniform synthetic aperture radiometer image reconstruction, which allows to achieve more accurate results compared to traditional methods.

The work is interesting, but requires significant improvement.

First of all, extensive proofreading is required.

I recommend clarifying the title of the paper. To determine the scope of the research, it would be useful to indicate that the work relates to radiometers.

In the abstract and introduction, I recommend starting with a problem statement so that the reader immediately understands the purpose of the work. In this case, it must be indicated that the work relates to radiometers. The authors start by looking at various aspects of antenna arrays.

In the abstract and in the introduction, it should be clearly stated what the authors have done new in their work (the novelty in described the paper, but rather vaguely).

My comments on the text of the paper:

On page 2 (line 45) the authors mention G-matrix reconstruction and FFT methods. Reference should be made to relevant works and at least a brief description of these methods should be given.

Figure 1. The authors provide a diagram, but do not explain what the designations are (U, V, Θ, Ä™, etc.)

Line 92. Please explain the abbreviation BT distribution.

Equation 3. What I and J are? (J is explained, but later in the text).

Line 117-118. What are S, F, P?

Line 120. Please explain the meaning of the term “local features”

Line 142-143. What is the reason for choosing such a number of antennas?

Figure 2 (left). What are the x and y axes? (please indicate in the figure)

Figure 6. Please specify the symbol for the y-axis.

The list of references contains publications from 2018-2019, but there are no works from 2020 and subsequent years. This topic is quite popular and therefore I recommend authors to include newer publications.